# Affect of Secondary Beam Non-Uniformity on Plasma Potential Measurements by HIBD with Split-Plate Detector

**DOI:** 10.3390/s22145135

**Published:** 2022-07-08

**Authors:** Igor Nedzelskiy, Artur Malaquias, Rafael Henriques, Ridhima Sharma

**Affiliations:** 1Instituto de Plasma e Fusão Nuclear, Instituto Superior Técnico, Universidade de Lisboa, Av. Rovisco Pais, 1049-001 Lisboa, Portugal; artur.malaquias@ipfn.tecnico.ulisboa.pt (A.M.); rhenriques@ipfn.tecnico.ulisboa.pt (R.H.); 2Culham Science Centre, Abingdon OX14 3DB, UK; ridhima.sharma@ukaea.uk

**Keywords:** nuclear fusion diagnostics, heavy ion beam probe, heavy ion beam diagnostic, plasma potential measurements, plasma poloidal magnetic field measurements

## Abstract

In a Heavy Ion Beam Diagnostic (HIBD), the plasma potential is obtained by measuring the energy of the secondary ions resulting from beam-plasma collisions by an electrostatic energy analyzer with split-plate detector (SPD), which relates the secondary ion beam energy variation to its position determined by the difference in currents between the split plates. Conventionally, the data from SPD are analyzed with the assumption that the secondary beam current is uniform. However, the secondary beam presents an effective projection of the primary beam, the current of which, as a rule, has a bell-like non-uniform profile. This paper presents: (i) the general features of the secondary beam profile formation, considered in the simplistic approximation of the circular primary beam and the secondary ions that emerge orthogonal to the primary beam axis, (ii) details of spit-plate detection and the influence of the secondary beam non-uniformity on plasma potential measurements, (iii) supported experimental data from the tokamak ISTTOK HIBD for primary and secondary beam profiles and the SPD transfer characteristic, obtained for the 90° cylindrical energy analyzer (90° CEA) and (iv) the implementation of a multiple cell array detector (MCAD) with dedicated resolution for the measurements of secondary beam profile and MCAD operation in multi-split-plate detection mode for direct measurements of the SPD transfer characteristic.

## 1. Introduction

The Heavy Ion Beam Diagnostic (HIBD) [1] is the only tool for direct measurements of plasma potential in magnetically confined fusion plasmas. The measurements of plasma potential fluctuations are of special interest and importance in investigations of turbulent transport [2]. Traditionally, the plasma potential is obtained from the measurements of secondary beam energy using an electrostatic energy analyzer with split-plate detector (SPD), which transforms the beam energy variation to the difference in currents between split plates [3]. Conventionally, with the assumption of relatively small electron density and temperature gradients inside the sample (primary beam ionization) volume, the data from SPD are analyzed with the assumption that the secondary beam current is uniform. However, because the secondary beam presents an effective projection of the primary beam inside the sample volume, it should be coupled with the profile of the primary beam, which, generally, is non-uniform.

This paper considers the possible influence of the secondary beam non-uniformity on plasma potential and its fluctuation measurements using SPD technique.

The paper is organized as follows. The coupling of primary and secondary beam profiles is examined in Section 2 for a simplistic model of circular primary beam and secondary ions that emerge perpendicular to the primary beam axis. Split-plate detection and SPD transfer characteristics for the non-uniform secondary beam are considered in Section 3. The influence of the secondary beam non-uniformity on plasma potential and its fluctuation measurements is analyzed in Section 4. The supported experimental data from the ISTTOK tokamak HIBD for primary and secondary beam profiles and SPD transfer characteristics obtained for the 90° cylindrical energy analyzer (90° CEA) are presented in Section 5. The multiple-cell and multi-split-plate detection for direct measurements of the secondary beam profile and SPD transfer characteristic is proposed in Section 6. Section 7 provides a summary.

## 2. Coupling of Primary and Secondary Beam Profiles

In HIBD, a beam of singly charged ions (primary beam, I^+^) is directed through the plasma across the confining magnetic field with energy to provide a Larmor radius higher than the dimension of the plasma cross-section, as shown in Figure 1a. Colliding with plasma electrons, some primary ions are ionized to the double-charge state (secondary beam, I^++^) along the primary beam path and are separated from the primary beam by the magnetic field. If to place a small aperture detector in the fan of the secondary ions that emerged from the plasma, it will only observe those secondaries that are created in a small sample volume along the primary beam, as schematically depicted in Figure 1c.

In Figure 1a, the injection angle of the primary beam is fixed and its trajectory inside the plasma presents an effective detector line of the measurements. In traditional HIBD shown in Figure 1b, the primary beam injection angle is scanned during plasma shot, and a fan of secondary ions created inside the definite sample volume of every angle discriminated primary beam trajectory is focused in one spatially definite point by the toroidal magnetic field. The respective sample volumes in the angle fan of the primary beam trajectories inside the plasma form the detector line of the measurements. In HIBD configuration in Figure 1a, the plasma parameters along the detector line are measured simultaneously with multichannel detection of the secondary ions. This HIBD configuration is used on the ISTTOK [4]. On the other hand, in Figure 1b showing traditional HIBD configuration with detection of the focused secondary ions in one spatial point, the measurements of the plasma parameters in the sample volumes along the detector line are delayed in time. In both HIBD configurations, the formation of the sample volume is similar (as in Figure 1c) with the sample volume size determined by the dimensions of the primary beam and detector aperture.

Consider the primary beam of circular cross section and uniform current density. The sample volume then consists of a cylinder with parallel end planes that cut the cylinder at some angle *θ* with respect to the cylindrical (beam) axis. In the YZ plane projection in Figure 1c, the sample volume cross-section is roughly a parallelogram. Diagonal of the parallelogram is minimal at *θ* = 90°, determining maximal spatial resolution of the measurements. For that hypothetical ideal situation, the 3D image of the sample volume of unit length (determined by detector slit width in the vertical Z direction) and circular primary beam of unit radius (in XY plane) is shown in Figure 2a with the emerged (in Y direction) secondary ions marked by arrows. As illustrated in Figure 2b the current distribution (profile) of the secondary beam is determined by the secondary ions created (integrated) along the chords in the XY cross-section of the primary beam. The respective 3D image of the secondary beam is shown in Figure 2c, demonstrating parabolic (*y* = 1 − *x*^2^) and rectangular shapes of the beam in XY and XZ planes, determined, respectively, by chords lengths and the analyzer entrance slit width.

These simplistic considerations demonstrate that even for the uniform primary beam, the secondary beam can attain non-uniformity as a result of the shape of the primary beam.

The 3D and 2D images of the above considered circular primary beam, but with parabolic, *y* = 1 − *x*^2^, and bell-shape, *y* = exp(−5*x*^2^), current density profiles are shown, respectively, in Figure 3a,b.

Figure 4a,b present the respective normalized 3D images and XY profiles of the secondary beam obtained in similar simplified considerations by integration in planes of primary beam chord slices, being:*y* = 1 − *x*^2^,*y* = 1 − 1.556*x*^2^ + 0.558*x*^4^*y* = exp(−5*x*^2^)(1)
for the, respectively, uniform (*y* = 1), parabolic (*y* = 1 − *x*^2^) and bell-shape (*y*= exp(−5*x*^2^)) primary beam.

The results in Figure 4 demonstrate the modifications of the secondary beam profile introduced by the non-uniformity of the primary beam. Additionally, notice that in real conditions of tangent propagation of secondary ions from the sample volume (as in Figure 1c), the respective XY cross-sections of the sample volume are rather ellipsoids than circles, with corresponding alterations in the secondary beam profile.

## 3. Split-Plate Detection and SPD Transfer Characteristics

In the HIBD standard split-plate detection, the secondary beam displacement, due to energy variation, is measured from the difference in the currents between the split plates. Conventionally, this can be explained according to the schematic of ideal case depicted in Figure 5a for the uniform rectangular beam of width *w_b_* [3]. From simple geometrical considerations, the relationship between beam displacement ∆*z* and split-plate current difference Δ*i* = *i*_1_ − *i*_2_ between plates is:∆*z*/*w_b_* = *h* = ∆*i*/*i**_Σ_* = (*i*_1_ − *i*_2_)/(*i*_1_ +*i*_2_) = *δ**i*,(2)
where *h* = ∆*z*/*w_b_* is the normalized on the beam width displacement, *i_Σ_* = *i*_1_ + *i*_2_ is the sum split-plates current and *δi* is the normalized beam current difference.

The respective ideal secondary beam *δi_I_*-*h* SPD transfer characteristic resulting from the effective scan of the beam across split-line is shown in Figure 5b and is linear inside the dynamic range of *δi* = ±1 and *h* = ±1.

Formally, the current on the split plates can be represented by triple integral ∫d*x* ∫d*y*∫d*z* taken on the secondary beam current profile over the appropriate functions and bounds of *x, y,* and *z*. The beam current difference between split plates expressed in terms of current on one of the plates (for example, *i*_2_) is:Δ*i* = *i*_1_ − *i*_2_ = *i_Σ_* − 2*i*_2_,*δi* = 1 − 2*i*_2_/*i_Σ_*.(3)

Then, for the secondary beams in Figure 4 and beam displacement in the Z direction (split-line parallel to X axis as shown in Figure 6a), the current *i*_2_ and *δi_Z_*-*h* transfer characteristic are:*i*_2_ =_0_∫*^h^* d*z*
_−1_∫^1^d*x* _0_∫*^f^*^(*x*)^ d*y* = *Ch*,*δi_Z_* = 1 − 2*Ch*/*i_Σ_*(4)
where *y* = *f*(*x*) is the secondary beam profile in the XY plane and *C* = _−1_∫^1^ *f*(*x*)d*x* = Const.

Contrary, for the beam displacement in the X direction (split-line parallel to Z axis as shown in Figure 6b), the respective current *i*_2_ and *δi_X_*-*h* transfer characteristic are:*i*_2_ =_−1_∫*^h^* d*x*_0_∫*^f^*^(*x*)^ d*y*_0_∫^1^d*z* = *F*(*h*)*δi_X_* = 1 − 2*F*(*h*)/*i_Σ_*,(5)
where *F*(*h*) = _−1_∫^h^ *f*(*x*)d*x*.

The results of Equations (4) and (5) demonstrate the orthotropy of the *δi*-*h* transfer characteristics for beam displacements in Z and X directions, or effective dependence on the orientation of the analyzer energy dispersion plane. As follows from Equation (4), for beam displacement in the Z direction and split-line parallel to the X axis (analyzer energy dispersion in YZ plane), the linearity of the *δiz*-*h* characteristic is persisted for any beam profile in the XY plane, determining the effective applicability of SPD data analysis based on the assumption of secondary ion beam’s uniformity. This result applies for the HIBDs with the orientation of the analyzer energy dispersion plane coinciding with a plane of secondary beam propagation. This is typical for all traditional HIBDs operated with the injection of the angle-scanned primary beam (Figure 1b) and the mirror-type 30° Proca-Green electrostatic energy analyzer [3]. On ISTTOK, the HIBD is operated with the fixed injection angle of the primary beam (Figure 1a) and employs the 90° cylindrical energy analyzer (90° CEA) due to its multichannel operation ability along the analyzer non-dispersive direction [5]. Figure 7 shows the schematic of the 90° CEA arrangement on ISTTOK. Multichannel operation along non-dispersive Z direction is achieved by the orientation of the analyzer energy dispersion XY plane orthogonal to the YZ plane of secondary beam propagation. In that analyzer orientation, the secondary beam displacement due to energy variation is effectively converted to X direction (as shown in the inset in Figure 7) in Figure 6, and, in accordance with Equation (5), the SPD *δi_X_*-*h* transfer characteristic should be sensitive to the secondary beam profile in the XY plane, violating the assumption of secondary beam uniformity in the analysis of the respective data from SPD.

Consider the secondary beams of the 3D images in Figure 4 obtained in the above simplified considerations. They are marked in relation to the primary beam profiles as U (uniform), P (parabolic) and B (bell-like).

In accordance with Equation (5) for beam displacement in the X direction and using Equation (1), the respective secondary beam *δi_X_*-*h* transfer characteristics shown in Figure 8 are:*δi_XU_* =2(*h* − (1/3)*h*^3^)/1.333,*δi_XP_* = 2[*h* − (1.556/3)*h*^3^ + (0.558/5)*h*^5^]/1.169,*δi_XB_* = erf(5^1/2^*h*).(6)

The results in Figure 8 demonstrate the deviation and non-linearity of the *δi_XU_*-*h*. *δi_XP_*-*h* and *δi_XB_*-*h* transfer characteristics inside the dynamic range in comparison with ideal case of linear *δi_I_*-*h* characteristics in Figure 5. Additionally, the following accompanying features should be pointed: (i) the effective narrowing of the dynamic range and (ii) magnification of the curves slope near *h* = 0. The last feature for the initially centered on split-line beam is equivalent to effectively increasing the sensitivity, which gives an advantage in the measurements of plasma potential fluctuations in such conditions.

Finally, it should be noticed here that investigations of the poloidal magnetic field and its fluctuations generated by the plasma current in standard HIBD applications with 30° Proca-Green energy analyzer are realized by the measurements of secondary beam displacement on SPD in a plane orthogonal to the analyzer energy dispersion plane [6,7]. These measurements are therefore sensitive to the secondary beam profile.

## 4. The Influence of the Secondary Beam Non-Uniformity on Plasma Potential and Its Fluctuations Measurements

For better a comparison, Figure 9 presents the respective differences in *δi_U_, δi_P_*, *δi_B_* (subscript X is omitted) and *δi_I_*: *δi_U_*_,*P*,*B*_ − *δi_I_* (i.e., the result for ideal case of uniform secondary beam in Figure 5 is subtracted from the results in Figure 8). The curves plotted in that way represent the deviations induced by beam profile in the measurements of plasma potential as a function of the secondary beam centerline position *h*. Note up to 40% deviation on *δi_B_* in the case of the bell-shape beam profile (B), if the center of the secondary beam is positioned at *h*_0_ = 0.4 inside the dynamic range window.

The plasma potential *Φ_pl_* in measurements with 90° CEA is obtained from relation [5]:(*Φ_pl_*/*E*_0_) = [(*w_b_*/2*R*_0_)/*C_E_*]*δi*,(7)
where *w_b_* is the beam width, *R*_0_ is the central radios of 90° CEA, *C_E_* is the energy dispersion coefficient and *E*_0_ is the beam energy.

For *w_b_* = 10 mm, *R*_0_ = 105 mm, *C_E_* = 1.56 [8] and from the data in Figure 9 for the bell-shape profile of the secondary beam at *h*_0_ = 0.4, the discrepancy in plasma potential value due to the non-linearity of *δi*-*h* characteristic is Δ(*Φ_pl_*/*E*_0_) = 10^−2^, or 200 V in absolute value for the ISTTOK HIBD Xe^+^ primary beam energy of *E*_0_ = 20 keV.

The non-linear response of the SPD on the secondary beam non-uniformity also influences the measurements of plasma potential fluctuations. Figure 10 shows the *δi* response on harmonic variation of the secondary beam position on SPD, given by:*h* = *h*_0_ + *a_h_*sin(*t*),(8)
for bell-shape (*δi_B_*, Figure 10a) and ideal uniform (*δi_I_*, Figure 10b) beams, *a_h_* = 0.1 and different *h*_0_.

Comparison of the results in Figure 10a for the bell-shape (*δi_B_*) beam and in Figure 10b for the ideal (*δi_I_*) uniform beam shows distortions (in amplitude and shape) in harmonic variation in the secondary beam position introduced by the non-linearity of the SPD response for different *h*_0_.

It should be pointed out that in the cross-correlation analysis of the plasma fluctuations realized in simultaneous multichannel measurements, the difference in the background plasma potential in sample volumes determines the different relative positions of the analyzed secondary beams on SPD (different operation points *h*_0_ on *δi*-*h* characteristic), attributing, therefore, a different response to the plasma potential fluctuations from different sample volumes.

## 5. Relevant Experimental Data from ISTTOK HIBD

### 5.1. Primary and Secondary Beam Profiles

The primary beam-line of the ISTTOK tokamak (*R*_0_ = 0.46 m, *a* = 0.085 m, *B* = 0.5 T, *I_p_* = 7 kA, *n_e_* =5 × 10^18^ m^−3^, *T_e_* = 100 eV) HIBD injector shown in Figure 11 [9] includes a set of electrostatic plates, wire monitor to control the primary beam profile in two directions (2 × 2 crossed in X and Y directions wires 0.3 mm of diameter) and defining slit (2 × 6 mm^2^ of dimensions in the respective X and Y directions).

The normalized profile of Xe^+^ primary ions obtained by scanning across the wires by electrostatic plates is shown in Figure 12 for two different background pressures (before and during plasma shot) in the primary beamline. It has a bell-like shape with full width at half maximum (FWHM) of FWHM = 1.5 mm at lower pressure, being widened at higher pressure due to the raising effect of beam scattering on the background gas particles.

Figure 13 presents the normalized profiles of the Xe^+^ beam observed in two (Y and X) directions on the primary beam multiple-cell detector (PBD) at the bottom of the tokamak chamber. The PBD presents a matrix of 15 cells (3 × 8 mm^2^ of dimensions in respective X and Y directions) arranged into 5 rows (in X direction) and 3 columns (in Y direction), as shown in the insets in Figure 13. Despite the scrape-off by defining aperture, the primary beam profiles on PBD are close to the bell shape. The FWHM of the Gaussians fitting the experimental points are, respectively, FWHM = 6 mm in X and FWHM = 8 mm in Y directions, indicating approximately ~6 mrad (0.34°) of primary beam divergence (the difference in FWHMs is related to the difference in the respective dimensions of defining slit and PBD cells).

Figure 14 shows the data obtained for the secondary Xe^++^ beam profile in the XY plane in Figure 7 by electrostatic scanning across the cell of 3 mm width incorporated into the “stop” detector in ISTTOK HIBD experiments with time-of-flight (TOF) measurements of plasma potential [10]. The experimental points indicate a peaked non uniform secondary beam profile with rather wide wings. The FWHM of the fitting Gaussian is FWHM = 6 mm.

### 5.2. δi-h Characteristic

Figure 15 shows the *δ**i*-*h* SPD transfer characteristics of 90° CEA obtained on ISTTOK for the secondary Xe^++^ beam in several plasma shots by shot-to-shot variation in the voltage on the analyzer plates for the analyzer operation in normal, Figure 15a, and 2-times deceleration, Figure 15b, modes [8]. Despite the scattering of the data due to not absolute repeatability of the plasma parameters from shot-to-shot, the *δ**i*-*h* characteristics clearly differ, being almost linear in normal mode and non-linear in deceleration mode. For comparison, the graphics of the ideal *δi_I_*–*h* and parabolic *δ**i_P_*-*h* transfer characteristics in Figure 8 are also shown, respectively, in Figure 15a,b.

In ISTTOK HIBD, the positions and dimensions of sample volumes are determined by the positions and dimensions of the cells of the multiple cell array detector (MCAD) [4]. The existing physical and geometrical constraints dictate the layout of the secondary beam propagation from MCAD location to the 90° CEA provided (in similarity as in HIBD TOF experiments [9]) by the cylindrical electrostatic plates incorporated on the back of the MCAD dedicated apertures and followed by a multi-aperture einzel lens [11]. Obviously, in that way, the profile of the secondary beam at SPD can undergo transformations due to the ion optics aberrations and apertures-slits clippings. Matching the secondary beam profile in Figure 14 with the 90° CEA entrance slit width of 5 mm, the cutting of the profile wings (blue vertical lines in Figure 14) and transformation to an effectively more uniform profile may be expected, thus explaining the observed almost linearity of the *δi*-*h* characteristic in Figure 15a. On the other hand, as shown in [12], the operation of the 90° CEA in deceleration mode is characterized by modifications to the beam shape (elongation of beam image in energy dispersion direction proportional to the deceleration coefficient) and heterogeneity imposed by aberrations introduced by the analyzer. Such a property of the 90° CEA operation in deceleration mode can be a reason for the observed *δi*-*h* characteristic non-linearity in Figure 15b.

## 6. Multiple-Cell and Multi-Split-Plate Detection

The above considerations indicate the necessity of the secondary beam profile control. In real experimental conditions, it is also important in standard HIBD applications for plasma potential and its fluctuations measurements due to effects of misaligned and scraped-off secondary beam as a result of mechanical interferences in the secondary beamline and analyzer entrance slits, aggravated by non-uniform components of the magnetic field along the beam propagation path [13,14]. The measurements of the secondary beam profile on SPD in one plasma shot are not trivial and to solve the issue, the implementation of MCAD with dedicated resolution may be suggested.

Figure 16 shows a number of MCAD cells of width *l_cell_* distributed over the SPD dynamic range window. With that MCAD there is a possibility to determine the secondary beam profile in the energy dispersion plane in one plasma shot with a spatial resolution of the cell width. Additionally, there is a possibility to measure directly the *δi*-*h* characteristic by successive subtraction of the acquired signals from the cells, given by:*δi_n_* = (_j=1_Σ^n^*i_j_* − _j=n+1_Σ^N^*i_j_*)/_i=1_Σ^N^*i_j_*.(9)

Such measurements are equivalent to effective scanning of *δi*-*h* characteristic, while MCAD operating in that mode may be considered as a multi-split-plate detector (MSPD).

The number and minimal width of the cells of MCAD in Figure 16 can be estimated from the analyzer dynamic range, (Δ*E*/*E*_0_)*_DR_*, and the minimum detectable change of ion beam energy, (Δ*E*/*E*_0_)*_min_*, determined by the signal-to-noise ratio (SNR) of the beam current measured on the cell. As an example, for the 90° CEA, the (Δ*E*/*E*_0_)*_DR_* and (Δ*E*/*E*_0_)*_min_* in terms of *δi* are [5]:(Δ*E*/*E*_0_)*_DR_* = [(*w_b_*/*R*_0_)/*C_E_*], (*δi* = ±1)(10)
(Δ*E*/*E*_0_)*_min_*= [(*w_b_*/2*R*_0_)/*C_E_*](SNR)^−1^, (*δi* = (SNR)^−1^).(11)

Estimation of cell number (and therefore the number of reliable points *N_est_*, or resolution of the *δi*-*h* characteristic measurements) is given by the ratio of (Δ*E*/*E*_0_)*_DR_*/(Δ*E*/*E*_0_)*_min_*, being from Equations (10) and (11) *N_est_* = 2(SNR). Then, estimation for the corresponding cell width is *l_cell_* = *w_b_*/*N_est_*, and, particularly for SNR = 10 and *w_b_* = 10 mm (for 90° CEA operated in 2-times deceleration mode [12]); this results in the minimal cell width of *l_cell_* = 0.5 mm.

## 7. Summary

The coupling of the primary and secondary beam profiles inside the sample volume is examined in the simplistic model of circular primary beam and secondary ions that emerge orthogonal to the primary beam axis. These considerations demonstrate that even for the uniform primary beam, the secondary beam can attain non-uniformity determined by the shape of the primary beam with additional modifications, if the primary beam is non-uniform. In the frame of that simplistic model, the *δi*-*h* transfer characteristics of the non-uniform secondary beam are orthotropic in directions of beam displacement depending on the orientation of the analyzer energy dispersion plane. In the case of energy dispersion and beam propagation planes coincidence, the linearity of *δi*-*h* characteristic is persisted for any secondary beam profile in a plane orthogonal to the plane of beam propagation. It is correct for practically all traditional HIBDs operated with angle-scanned primary beam and the mirror-type 30° Proca-Green electrostatic energy analyzer. In the case of energy dispersion and the beam propagation planes orthogonality, the SPD *δi*-*h* transfer characteristic is non-linear, determined by the respective secondary beam profile. This situation takes place for ISTTOK HIBD operated with fixed injection angle primary beam and 90° CEA. The non-linearity of *δi*-*h* transfer characteristic was observed experimentally for 90° CEA operated in 2-times deceleration mode. In that case, the assumption of secondary beam uniformity in the analysis of the respective data from SPD is violated and requires knowledge of the secondary beam profile. The corresponding measurements can be realized with MCAD of dedicated resolution. In addition, for that MCAD operated in multiple-split-plate configuration, the direct measurements of *δi*-*h* transfer characteristic are available. Additionally, for the non-uniform profiles of the secondary beam, the MCAD data can be used as a supplement for plasma potential measurements by evaluation the shift in center of mass of the secondary beam current distribution.

Finally, it should be noted that knowledge of the secondary beam profile and its influence on the measurements is also important in traditional HIBD applications in investigations of the poloidal magnetic field and its fluctuations generated by plasma current, and if the secondary beam misalignment and scrape-off are suspected.

## Figures and Tables

**Figure 1 sensors-22-05135-f001:**
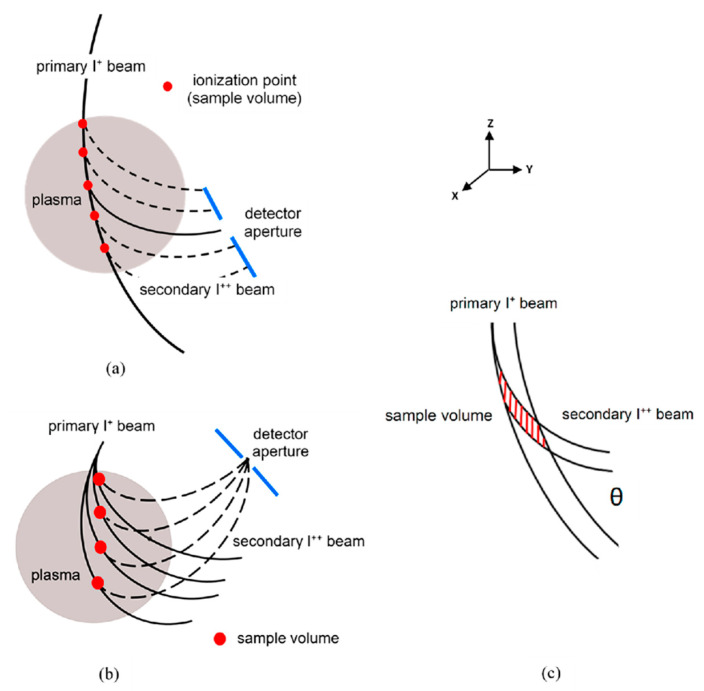
Schematics of HIBD primary and secondary beam trajectories (**a**,**b**) and sample volume formation (**c**).

**Figure 2 sensors-22-05135-f002:**
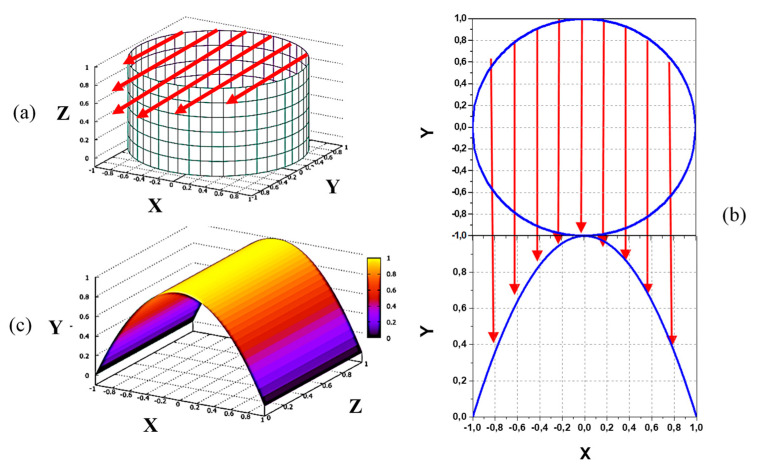
3D image of sample volume of unit length and circular primary beam of unit radius (**a**), formation of the secondary beam profile in XY plane (**b**), and respective 3D image of the secondary beam (**c**).

**Figure 3 sensors-22-05135-f003:**
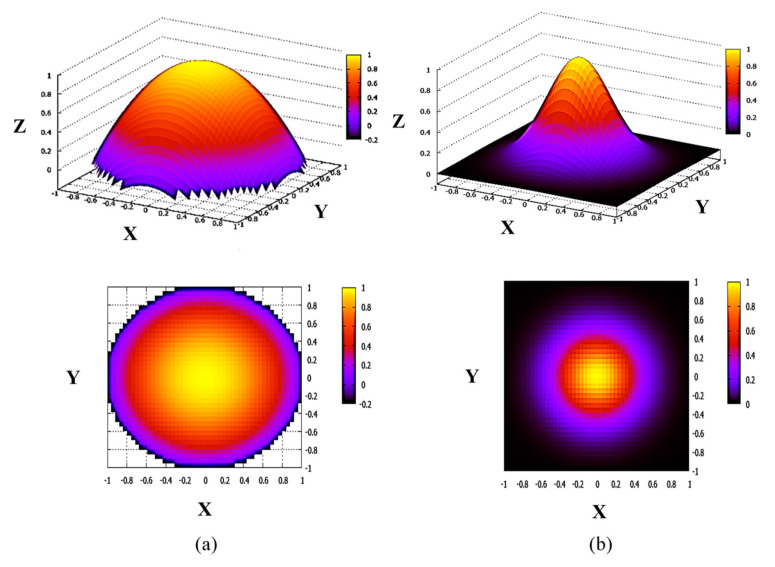
3D and 2D images of circular primary beam with parabolic, *y* = 1 − *x*^2^ (**a**) and bell-shape, *y* = exp(−5*x*^2^) (**b**), current density profiles.

**Figure 4 sensors-22-05135-f004:**
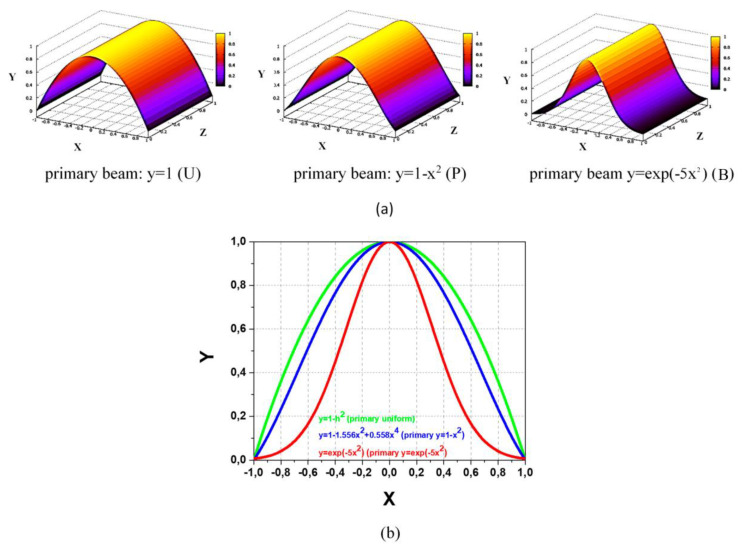
Normalized 3D images (**a**) and 2D profiles (**b**) of the secondary beam obtained by integration in planes of primary beam chord slices for uniform (*y* = 1), parabolic (*y* = 1 − *x*^2^) and bell-shape (*y*= exp(−5*x*^2^)) primary beam.

**Figure 5 sensors-22-05135-f005:**
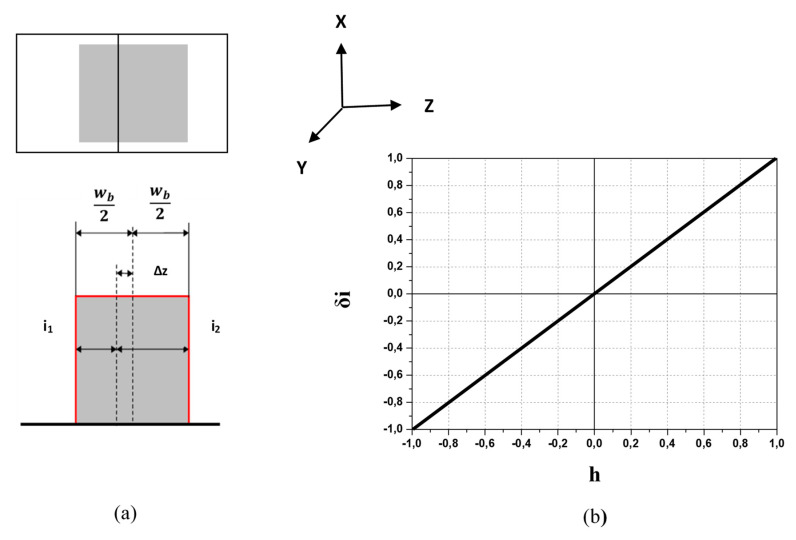
Schematic of split-plate detection for ideal case of uniform rectangular beam of width *w_b_* (**a**) and respective *δi_I_*-*h* SPD transfer characteristic inside the dynamic range of *δi* = ±1 and *h* = ±1 (**b**).

**Figure 6 sensors-22-05135-f006:**
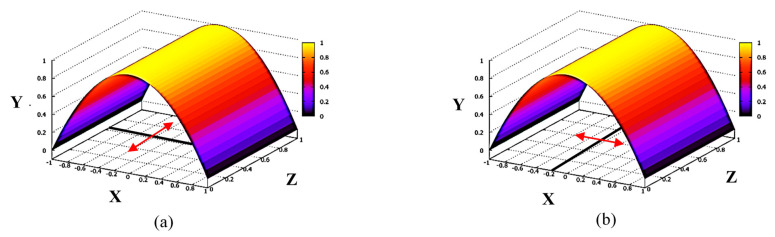
Orientation of split-line and secondary beam displacement: (**a**) beam displacement in Z direction with split-line parallel to X axis, and (**b**) beam displacement in X direction with split-line parallel to Z axis.

**Figure 7 sensors-22-05135-f007:**
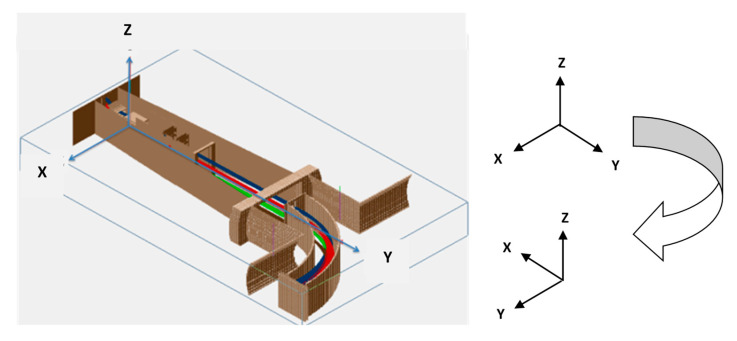
Schematic of the 90° CEA arrangement on ISTTOK.

**Figure 8 sensors-22-05135-f008:**
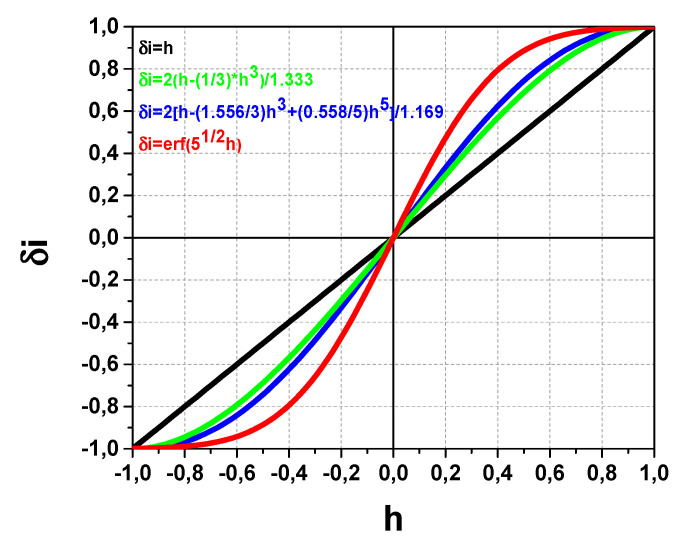
*δi_X_*-*h* transfer characteristics for uniform (U), parabolic (P) and bell-like (B) profiles.

**Figure 9 sensors-22-05135-f009:**
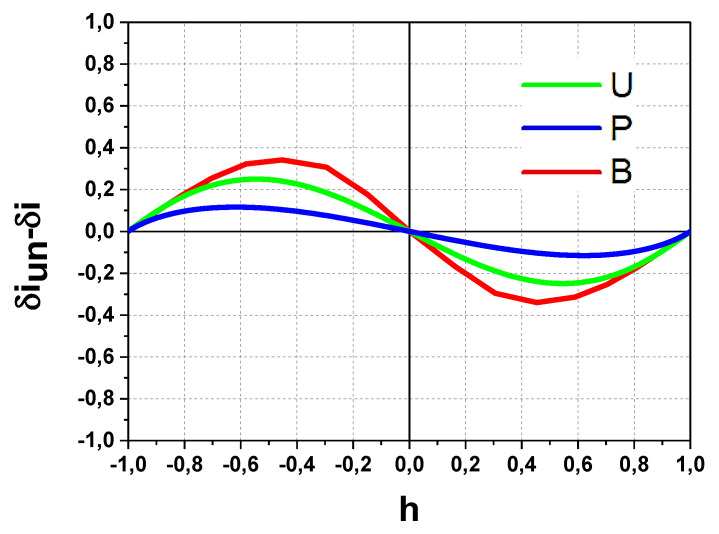
Deviations induced by the beam profile in measurements of plasma potential as a function of the secondary beam centerline position *h* for uniform (U), parabolic (P) and bell-shape (B) beam profile.

**Figure 10 sensors-22-05135-f010:**
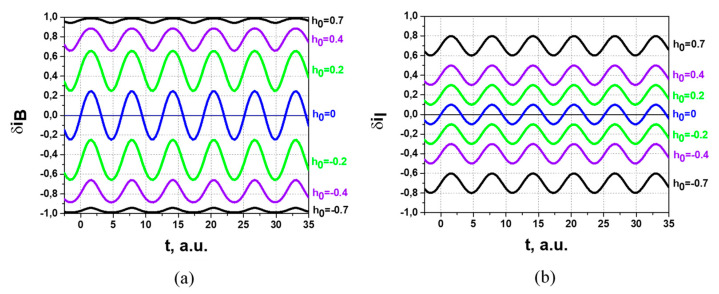
*δi* response on harmonic variation of beam position on SPD for different *h*_0_ for bell-shape (**a**) and ideal uniform (**b**) beam.

**Figure 11 sensors-22-05135-f011:**
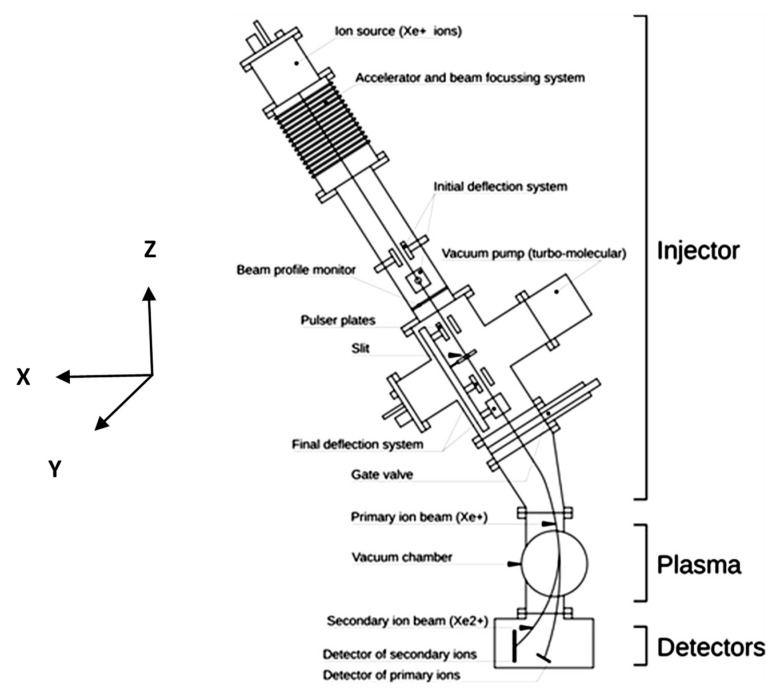
Primary beamline of ISTTOK HIBD injector.

**Figure 12 sensors-22-05135-f012:**
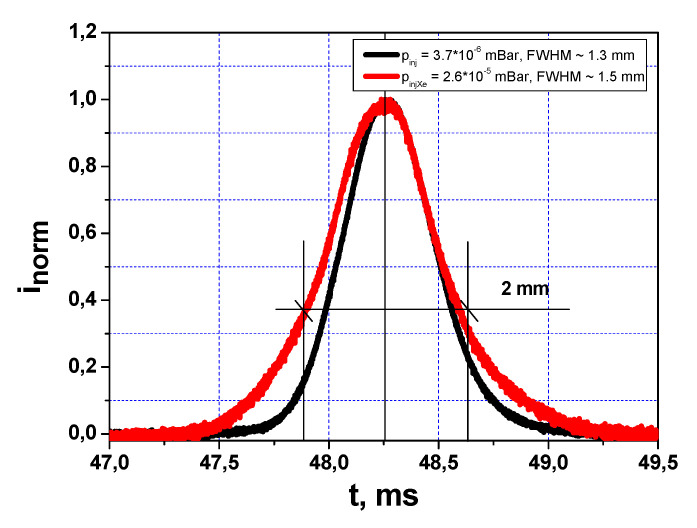
Normalized profiles of Xe^+^ primary ions across wire monitor.

**Figure 13 sensors-22-05135-f013:**
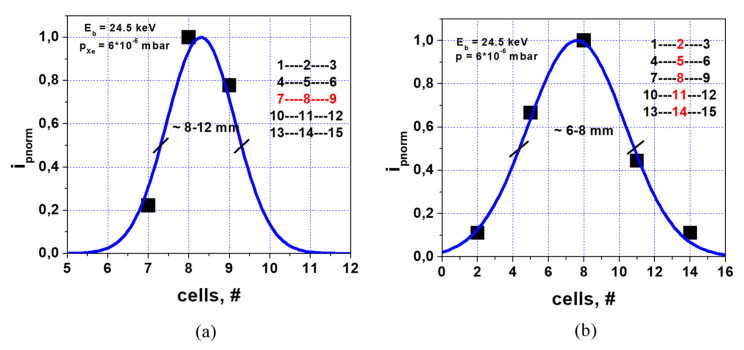
Normalized profiles of Xe^+^ primary ions on primary beam detector in Y (**a**) and X (**b**) directions.

**Figure 14 sensors-22-05135-f014:**
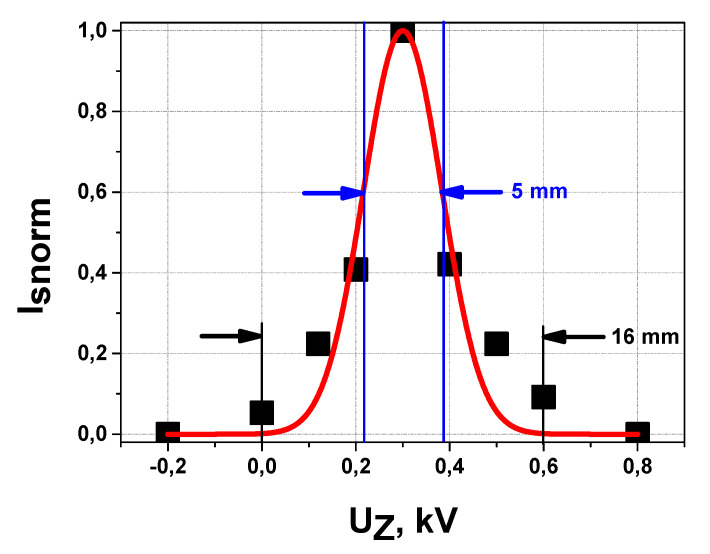
Secondary Xe^++^ beam profile on “stop” detector in ISTTOK HIBD experiments with time-of-flight measurements of plasma potential.

**Figure 15 sensors-22-05135-f015:**
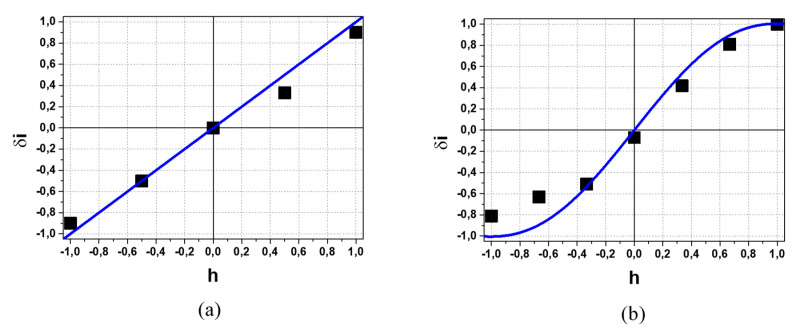
*δi*-*h* transfer characteristics of the secondary Xe^++^ beam for 90° CEA operation in normal (**a**) and 2-times deceleration (**b**) mode.

**Figure 16 sensors-22-05135-f016:**
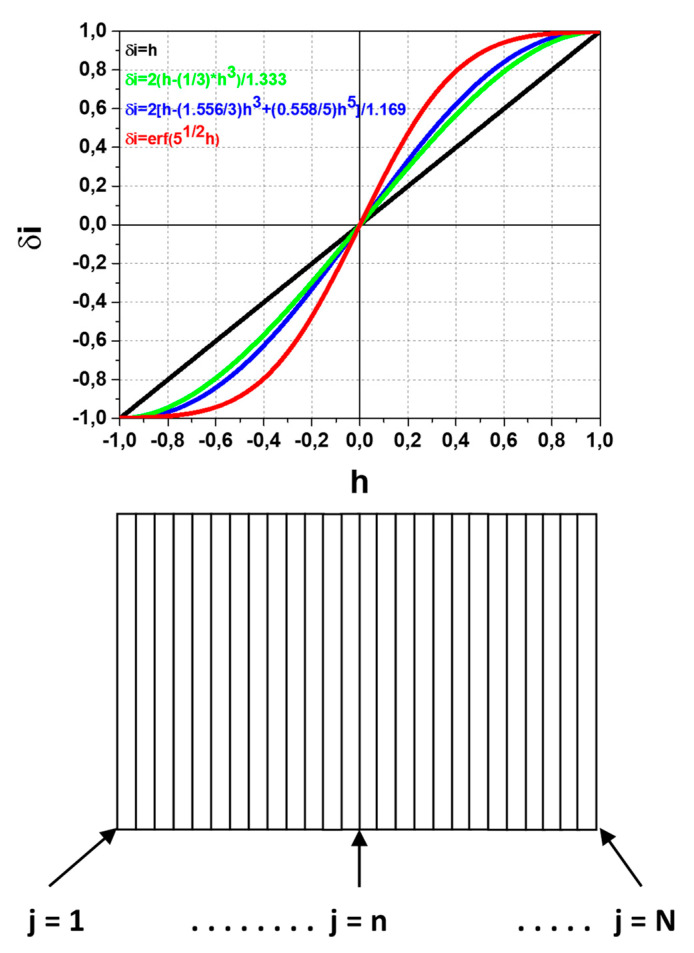
Schematic of multiple cell array detector for secondary beam profile and *δi*-*h* transfer characteristic measurements.

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
