# Peer review of "Affect of Secondary Beam Non-Uniformity on Plasma Potential Measurements by HIBD with Split-Plate Detector"

_sensors, 2022, doi:10.3390/s22145135_

Round 1
Reviewer 1 Report
many figures in the paper is not clear, I suggest the authors to change the figures to the vector format. This will make them clear in the pdf file. The figures are very important to show the results in a research paper.
Author Response
We are grateful to reviewer for suggestion and improved all the figures.
Reviewer 2 Report
Nedzelskiy et al. investigated the influence of secondary beam non-uniformity on plasma potential measurements by heavy ion beam diagnostic (HIBD) with split-plate detector (SPD). They examined the coupling of primary and secondary beam profiles for simplistic model of circular primary beam and secondary ions emerged perpendicular to the primary beam axis. Secondly, they discussed the influence of the secondary beam non-uniformity on plasma potential and its fluctuations measurements. Finally, the multiple-cell and multi-split-plate detection for direct measurements of the secondary beam profile and SPD transfer characteristic is proposed. Overall, they provided a lot of information, and I think the manuscript may be acceptable after minor revision. Figure 1 is not clear, please replace the figure with a higher resolution and DPI. For other figures, the scale and text fonts are too small, please use a larger font.
Author Response
We are grateful to reviewer for the positive assessment of our work. In accordance with suggestion all the figures are improved.
Reviewer 3 Report
The manuscript entitled “Affect of secondary beam non-uniformity on plasma potential measurements by HIBD with split-plate detector” describes a probe beam diagnostics with heavy-ion beams such as from Xenon sources that are used for measuring plasma potential and density in tokamaks. Different profiles of the primary beam (uniform, parabolic, and gaussian) are evaluated with respect to the profile of the secondary beam. Then, experimental data from ISTTOK is shown for primary and secondary beam. At last, a multi-split plate detector is modeled to enable single-shot measurements.
I have only minor points on the manuscript that could help to improve the quality:
-
About two of three citations are self-citations of the authors. Including more citations from other groups would support the importance of the detection system for tokamaks.
-
Improving the resolution of Fig. 2, 3, 4, 6, 8, and 15 would enable the reader to read the text in the images.
-
For Fig. 11, it would be helpful to indicate the size of the setup, e.g. radius of the plasma vacuum camber.
-
How is this diagnostics deployed at other tokamaks? Can there be references added from other groups?
This is a good work on the topic. The manuscript is suitable for publication in Sensors in the current version.
Author Response
We are grateful to reviewer for the positive assessment of our work.
Reply to the comments:
- To our knowledge, there were not any detail considerations of secondary beam profile and its influence on plasma potential measurements by HIBD with SPD. The influence of misalignment and scrape-off on secondary beam profile and resulted problems of beam energy measurements are just mentioned in Ref. 13 and Ref. 14.
- All the figures are improved.
- Parameters of the ISTTOK tokamak are added at the beginning of Subchapter 5.1.
- The review and details of HIBD employment on tokamaks and stellarators can be found in Ref.2
Round 2
Reviewer 1 Report
now it can be accepted.